# Histopathology of Budd–Chiari Syndrome

**DOI:** 10.3390/diagnostics13152487

**Published:** 2023-07-26

**Authors:** Alberto Quaglia

**Affiliations:** 1Department of Cellular Pathology, Royal Free London NHS Foundation Trust, Pond Street, London NW3 2QG, UK; a.quaglia@ucl.ac.uk; 2UCL Cancer Institute, London WC1E 6DD, UK

**Keywords:** Budd–Chiari syndrome, sinusoidal dilatation, congestion, perisinusoidal fibrosis, nodular regenerative hyperplasia, parenchymal extinction, cirrhosis, venous outflow obstruction

## Abstract

The histopathological changes in Budd–Chiari syndrome (BCS) overlap with those of sinusoidal obstruction syndrome (SOS) and of cardiac or pericardiac disorders resulting in right cardiac failure. These conditions, however, are different on both clinical and pathological grounds and need to be differentiated from BCS. This review is centred on the three main aspects of BCS in diagnostic liver histopathology: (1) general histopathology of BCS; (2) implications for liver biopsy interpretation; and (3) BCS in the liver allograft. The histological features of BCS form a complex spectrum which is shaped differently in each individual case according to the topographical distribution and chronological evolution of the obliterative insult, its upstream effect of the hepatic vascularisation and the consequent parenchymal injury, scarring and remodelling. Sampling variation limits the use of liver biopsy for prognostication in patients with BCS.

## 1. Introduction

The pathogenetic classification of Budd–Chiari syndrome is critical to the understanding of the histopathological aspects of this disease. Hepatic venous outflow obstruction, characterised by an obstacle to the hepatic veins or suprahepatic portion of the inferior vena cava (IVC) at any level between small intrahepatic veins and the right atrium [1], is considered primary when caused by a primary vascular disorder and secondary when caused by a non-vascular process such as a tumour compressing or infiltrating the hepatic veins or inferior vena cava (IVC). The histopathological changes in Budd–Chiari syndrome (BCS) overlap with those of sinusoidal obstruction syndrome (SOS) and of cardiac or pericardiac disorders resulting in right cardiac failure. These conditions, however, are different on both clinical and pathological grounds and need to be differentiated from BCS [1].

The histopathological features of BCS can be roughly divided into two main groups: (1) changes related to venous outflow obstruction and (2) changes related to the underlying cause. The causes of BCS are discussed in Section 3 of this edition. The general principle is that primary vascular causes of BCS revolve around the concept of the Virchow triad, namely, vascular injury, venous stasis and hypercoagulable states. This review is divided into the following subsections: (2) general histopathology of BCS; (3) implications for liver biopsy interpretation; and (4) BCS in the context of liver allograft.

## 2. General Histopathology of BCS

The primary histological manifestations of BCS derive from complete or partial obstruction of the hepatic outflow, resulting in upstream venous stasis. Hepatic blood flow is protected by a number of compensatory mechanisms which include increased arterial flow, increased portal pressure and redistribution of the portal blood to areas where the outflow is still preserved as well as the development of a collateral hepatic venous circulation [1]. In primary vascular disorders, obliteration of the hepatic veins can be in the form of a fresh or organised thrombus followed by the development of local obliterative and bridging fibrosis over time. The thrombotic events are distributed heterogeneously in the liver and are not synchronous. Obliterated and patent hepatic veins and newly organised and recurrent thrombi often co-exist, causing variable alterations of the venous flow in different regions. The appearance of thrombi depends on their age. The lumen of an acutely thrombosed hepatic vein fills with granulation tissue and fibrosis. Fibrosis leads to retraction and reduction of the vein size [2]. As recanalization develops, the vein shows intimal fibrosis, intraluminal delicate webs and intraluminal channels. The intimal collagen can be layered, indicating recurrent thrombotic events [2]. Small hepatic veins, located upstream of thrombosed large veins, show mural oedema, congestion and fibrosis but no thrombus [2].

Partial or complete blockage of a hepatic vein causes dilatation with or without congestion of perivenular and mid-lobular sinusoids and a local hypoxic state. There may be extravasation of red blood cells into the space of Disse. Hepatic plates are variably affected, showing a spectrum of changes. The milder forms are characterised by hepatocellular atrophy and mild sinusoidal dilatation [2] and thinning of the hepatic plates. Frank hepatocellular necrosis, at times of the coagulative type, can also occur and is probably related to concomitant portal vein thrombosis [3,4].

Hepatocytes in the centrilobular region can undergo changes probably representing a form of biliary metaplasia and including expression of cytokeratin 7 or even the formation of ductular structures. These changes are often associated with remodelling of the local microvasculature including the capillarization of sinusoids demonstrated with CD34 immunohistochemistry and microvessel formation including the ingrowth of well-formed arterioles as part of the fibrosing process [5]. Reduced glutamine synthetase expression in perivenular hepatocytes, along with periportal expression in some instances, suggests alterations in metabolic zonation or even reversed zonation. As a result, centrilobular changes can mimic very closely portal tracts and in turn the histological manifestations of biliary disorders [5]. The concept of ‘centrizonal injury disease’ has been proposed to capture the similarities between hepatic venous obstruction, alcohol-related liver disease (ARLD) and non-alcoholic fatty liver disease (NAFLD) [6]. Atrophic hepatocytes may contain abundant lipofuscin and small steatotic droplets, and canalicular cholestasis can be present along with haemosiderin-laden Kupffer cells [7]. Intracytoplasmic dPAS-positive globules have been observed in up to 8% of cases of venous outflow obstruction. These globules cross-react with alpha-1-antitrypsin (A1AT) globules [8] and are also observed in the livers of patients with right cardiac failure [9].

Confluent hepatocyte loss, and later fibrosis, can result in the linking of adjacent centrilobular regions and reverse lobulation (Figure 1). Fibrous areas may contain haemosiderin [10]. The histological manifestations of BCS are not confined to the centro-midlobular regions. Venous stasis can trigger thrombotic portal events, which can cause panlobular or multilobular atrophy, necrosis or extinction and the formation of parenchymal extinction lesions [2]. Periportal hepatocytes tend to show compensatory regenerative changes in the form of increases in the breadth of hepatic plates, with portal veins providing an alternative outflow tract to these regions [2], and which can evolve into frank nodular regenerative hyperplasia (NHR). Secondary portal vein obstruction can therefore alter the central and reverse lobulation pattern to a more veno-portal distribution, mimicking, therefore, other forms of chronic liver injury [2], as described in the seminal work by Tanaka and Wanless [4]. Depending on the co-existence and balance between hepatic and portal vein obstruction, four main parenchymal patterns of injury can develop: veno-centric cirrhosis, veno-portal cirrhosis, total parenchymal extinction and nodular regenerative hyperplasia. These different patterns can co-exist in different regions of the same liver [4], a critical point to consider when interpreting liver biopsy specimens. Portal changes mimicking biliary disease and including ductular reaction, portal inflammation and portal-based fibrosis are also described in venous outflow obstruction, even in the absence of concomitant biliary disease [11].

The complexity of the BCS changes often reaches a macroscopic scale and becomes apparent upon imaging or at the examination of surgically resected or post-mortem specimens. The effects of venous outflow block alternate with those of regeneration, creating sub-segmental, segmental or even lobar regions of parenchymal atrophy, extinction, fibrosis or regeneration. A notable feature of BCS is the hypertrophy of the caudate lobe due to its direct drainage into the IVC, bypassing blocked hepatic veins. Dominant parenchymal nodules, fed by compensatory arterial inflow [1], become visible macroscopically and need to be differentiated clinically and histologically from the focal-nodular-hyperplasia (FNH)-like, hepatocellular adenoma (HCA)-like and hepatocellular carcinomas (HCC) which can complicate BCS [12].

## 3. Implications for Liver Biopsy Interpretation

Liver biopsies from patients with BCS are infrequent and often deployed to target nodular lesions. EASL guidelines recommend the use of liver biopsy when the diagnosis is uncertain due to imaging failing to identify obstruction of large veins [13]. The term ‘Small hepatic veins Budd Chiari’ denotes a form of BCS in which the large hepatic veins are not affected. It has been described in the context of primary or secondary antiphospholipid syndrome (APS) or paroxysmal nocturnal haemoglobinuria [14].

The questions for the histopathologist are usually the following: Does this liver biopsy show features of venous outflow obstruction? Are the changes in venous outflow obstruction due to BCS? Is it acute or chronic BCS, and how severe is the liver injury?

### 3.1. Does This Liver Biopsy Show Features of Venous Outflow Obstruction?

Histological confirmation of venous outflow obstruction needs to be based on a set of strict criteria to avoid misdiagnosis, in terms of underestimating or overestimating histological signs of a vascular abnormality. Sinusoidal dilatation in isolation and focal in the biopsy specimen is usually non-specific. When confined to the biopsy, the core edge of the biopsy core, it is probably a fixation artefact. Sinusoidal breadth depends also on the state of the patient’s circulation at the time of biopsy [9]. Sinusoidal dilatation needs to be present consistently throughout the biopsy core to be considered significant. It may or may not be associated with sinusoidal congestion, hepatic plate atrophy and/or spill-over of red blood cells in the space of Disse. Erythrocytes may be washed away by processing, and congestion may not be present in tissue sections [10]. The sinusoidal dilatation of sickle cell disease is characterised by packing of the sinusoids by sickle erythrocytes [15]. Perisinusoidal fibrosis develops over time and may be variable in distribution.

Consistent significant sinusoidal dilatation (SD), defined as ‘a sinusoidal lumen that is more than one liver cell plate wide, observed in several lobules in a high-quality liver specimen devoid of artifactual tearing’ [16], is not specific to venous outflow obstruction or indicative of a specific primary vascular abnormality. Its significance needs to be considered based on the clinical context and biopsy indication [16]. Sinusoidal dilatation is considered to be non-obstructive when not explained by (1) vascular infiltration by sickle cells, hemophagocytic histiocytes or neoplastic cells causing enlargement of sinusoids; (2) BCS or heart failure; (3) small-for-size syndrome after liver transplantation; or (4) SOS/VOD [16]. Marzano et al. [17] have proposed three main hypotheses on the pathogenesis of non-obstructive sinusoidal dilatation: (1) atrophy of hepatocytes; (2) haemodynamic changes related to altered arterial and portal inflow; and (3) soluble systemic factors including interleukin-6 and VEGF as part of systemic inflammatory response syndrome (SIRS). These hypotheses fit with the observations of sinusoidal dilatation in association with a range of conditions including Hodgkin disease [15], solid tumours, systemic granulomatous disorders, anticardiolipin antibodies, antiphospholipid syndrome, hemophagocytic syndrome, in heroin addicts and in wedge biopsies taken at surgery [9]. It has been described historically in association with toxic agents thought to cause angiosarcoma (e.g., arsenic, vinyl chloride, and thorium dioxide (thorotrast)) [15]. Increased sinusoidal pressure resulting in sinusoidal dilatation can be due to increased portal pressure or increased arterial flow [18]. Obstruction of the portal vein or partial obstruction of the hepatic artery can cause haemorrhagic necrosis or atrophy of the centrilobular hepatic plates, causing dilatation of the sinusoids, which fill with blood, mimicking therefore venous outflow obstruction [10]. The historical term ‘infarct of Zahn’ refers to focal liver lesions visible macroscopically as sharply demarcated dark red areas corresponding histologically to dilated and severely congested often centrilobular sinusoids with or without associated hepatic plate atrophy, congestion and/or hepatocellular necrosis and nearby thrombotic portal vein occlusion [19]. The lobular distribution of sinusoidal dilatation may suggest an underlying abnormality or risk factor, but a specific zonal distribution does not imply necessarily a specific underlying primary vascular abnormality. Centrilobular sinusoidal dilatation can be present even without venous outflow block [16]. Periportal/midzonal sinusoidal dilatation can be associated with portal inflow abnormalities and shunting but is also observed in association with oral contraceptives. In patients with eclampsia, intrasinusoidal fibrin thrombi and signs of ischaemia are also present [15]. Fibrin thrombi in portal and sinusoidal vessels are also observed in diffuse intravascular coagulation. In nodular regenerative hyperplasia, sinusoidal dilatation may alternate with nodular hepatic plates often in a midzonal location. Midzonal dilatation has been described in patients with renal cell carcinoma [9]. Dilated sinusoids at times mimicking cavernous haemangioma along with dilated veins and arteries, vascular malformations and shunting, variable degrees of fibrosis, ischaemic cholangiopathy, NRH and focal nodular hyperplasia constitute the spectrum of changes in the livers of patients with haemorrhagic telangiectasia (Rendu–Osler–Weber). Liver biopsy is contraindicated in these patients due to the high risk of bleeding. Sinusoidal dilatation could also represent a secondary effect of a nearby space-occupying lesion or even an intralesional abnormality, an aspect to consider when dealing with hepatocellular lesions such as inflammatory adenoma which can blend with or mimic background liver tissues.

Red blood cells are normally present inside sinusoids, and there is no specific abnormal threshold without other accompanying pathological features. As mentioned earlier, red blood cells may be washed away during processing, so their absence in the presence of sinusoidal dilatation is of no significance [10]. In contrast, intrasinusoidal congestion confirms that sinusoidal dilatation was present before the biopsy specimen was taken [10]. In addition to venous outflow obstruction, sinusoidal congestion is observed in sickle cell disease and periportally in eclampsia. Erythrophagocytosis is unusual and usually observed in the context of haemophagocytic syndromes [9]. A variable degree of sinusoidal congestion or even extravasated red blood cells and centrilobular haemorrhage are often present in livers with massive hepatic necrosis [20].

Perisinusoidal fibrosis is not exclusive to venous outflow obstruction and can be observed in a variety of other conditions. It is a characteristic feature, for example, of steatohepatitis in which the fibrosis of the space of Disse is typically pericellular in a ‘chicken wire’ pattern rather than the more linear pattern of hepatic venous outflow obstruction. A more linear pattern is, however, observed after the resolution of steatohepatitis—for example, after alcohol abstinence [21]. Perisinusoidal fibrosis can also be observed as a result or centrilobular inflammation (e.g., autoimmune hepatitis), randomly in the lobule in diabetic hepatosclerosis or in association with hypervitaminosis A. In the liver transplant setting, perisinusoidal fibrosis tends to be observed as a result of various forms of rejection [9].

The term ‘peliosis hepatis’ refers to the presence of variably sized and randomly distributed [16] pools of blood not lined by endothelial cells, or with an incomplete endothelial lining [9], and containing preserved or degenerated erythrocytes or empty if erythrocytes have been washed away. They communicate with the sinusoidal lumen [18] and are probably caused by rupture of the sinusoidal wall [18] due to its weakening or necrosis of hepatocytes [18]. Peliosis hepatis is associated with androgen steroids and other drugs such as progesterone and azathioprine, as well as various other conditions including asphyxia and wasting disorders [9]. It can be seen inside hepatocellular lesions. Peliotic areas are usually distributed irregularly without a specific zonation, and nearby sinusoids may not be dilated, in contrast to venous outflow obstruction. Peliotic areas do not contain reticulin. Bacillary angiomatosis is caused by *Bartonella* spp. in AIDS patients or other immunocompromised patients [18]. Angiomatous lesions consisting of cystic-like spaces containing erythrocytes and lined by leukemic cells have been described in the portal tracts and in the lobules of livers of patients with hairy cell leukaemia [22]. Fused steatotic vacuoles as seen in lipopeliosis can mimic dilated sinusoids [18].

### 3.2. Are the Changes of Venous Outflow Obstruction Due to BCS?

As mentioned earlier, the histopathological changes of BCS overlap considerably with those of SOS and right cardiac failure, and these three conditions may be indistinguishable histologically, hence the preferred terminology of venous outflow obstruction, particularly when assessing liver biopsies [9]. SOS is caused by non-thrombotic obstruction of small (<300 microns in diameter) hepatic venules in a process that appears to start at the sinusoidal level with damage to and activation of sinusoidal endothelial cells; leakage of leukocytes, red blood cells and cell debris in the space of Disse; hepatocellular necrosis and subintimal oedema; and haemorrhage of the hepatic venules [18]. Intrasinusoidal thrombi can form later along with perisinusoidal fibrosis and eventually fibrous obliteration of hepatic venules characterised by concentric or eccentric intimal fibrosis and recanalization. Secondary thrombotic events of large hepatic veins can occur in SOS, and secondary obliterative change of small hepatic venules can occur in BCS, making their distinction impossible at times [18]. Some features may help in gauging differential diagnosis but are not diagnostic. Changes due to BCS are usually distributed heterogeneously in contrast to congestive hepatopathy, which is more uniform [23]. Hepatic veins and venules tend to be dilated in right cardiac failure, whilst hepatic venules can be attenuated, often narrowed and at times obliterated, containing reticulin strands, in SOS. Congestion is particularly prominent in SOS, and patchiness in a liver biopsy specimen may favour SOS rather than BCS or right heart failure, with congested areas alternating with areas with minimal changes and patent small hepatic venules being present [10]. Fibrous obliteration of hepatic venules can occur in BCS and right heart failure over time, so it is not a differential criterion [10]. In terms of other conditions, hepatocellular necrosis can occur in BCS and in acute cardiac failure due to, for example, pulmonary embolism and associated with plate haemorrhage or with shock; it is potentially problematic therefore to differentiate this condition from purely ischaemic lesions, which are often characterised by necrosis of individual hepatocytes or small clusters [10]. Right cardiac failure is usually confirmed clinically. SOS may be a diagnosis of exclusion based on clinico-pathological correlation. Congestion is the predominant finding of sickle cells disease and HLH [23]. Venous outflow block is a known complication of polycystic liver disease [24].

Features of venous outflow obstruction can be observed in the vicinity of space-occupying lesions, often along with portal changes of biliary obstruction. The clinical history usually is of a targeted liver biopsy, but the possibility of a space-occupying lesion should always be considered when dealing with unexpected features of venous outflow block in a liver biopsy specimen. Finally, vascular neoplasms can simulate features of venous outflow obstruction. Epithelioid haemangioendothelioma in particular is a known histological mimicker due to the invasion of hepatic venules and local sclerosis in the shape of organised vascular thrombi [18]. The appearance on imaging of angiosarcoma can be non-specific and resemble venous outflow obstruction.

### 3.3. Is It Acute or Chronic BCS, and How Severe Is Liver Injury?

Changes in the acute phase of BCS depend on the level and degree, distribution and duration of obstruction. Acute BCS can present with a clinical picture of acute liver failure. The diagnosis is usually reached clinically because hepatomegaly (with or without ascites) is not a common feature of acute liver failure, and liver biopsy is usually not necessary except in cases when there is diagnostic uncertainty [25]. Biopsies from 6 of 19 patients with BCS presenting as acute liver failure showed centrilobular congestion and hepatocellular necrosis, but no significant fibrosis [26]. Obstruction of the three major hepatic veins results in severe congestion with replacement of the hepatic plates by blood and the clinical presentation of painful hepatomegaly, ascites and liver dysfunction [2]. The involvement of one or two of the two main hepatic veins results in fewer or no symptoms [2,27,28]. Livers with such obstruction are enlarged, appear congested macroscopically and show round edges [18]. The sinusoids at the border of the haemorrhagic areas are usually periportal and dilated. Small hepatic venules may be dilated or narrowed and obliterated, depending on the underlying cause. Portal vein thrombosis may be present. Portal tracts tend to be normal or show biliary-type changes [9]. Acute clinical presentation of BCS is usually due to recent obstruction of a hepatic vein complicating chronic hepatic vein obstruction rather than synchronous acute onset disease [1]. The chronic setting is characterised essentially by the triad of hepatic (and often portal) vein changes, fibrosis and parenchymal remodelling. As mentioned earlier, the combination of primary centrilobular and secondary portal events results in a complex, evolving and topographically heterogeneous histological picture. Macroscopically, BCS livers can be shrunken [23] and show areas of segmental and/or lobal atrophy or scarring alternating with regions of regenerative hyperplasia where blood perfusion and drainage are preserved, and in particular caudate lobe hyperplasia. Venous thrombi are at various stages, which explains the episodic nature of symptoms related to the extension and recurrence of thrombosis [2], and thrombosed portal vein branches are often present. It is therefore possible that a liver biopsy shows exclusively regenerative changes with no obvious signs of venous outflow obstruction and that fibrosis and its severity may be underestimated or overestimated depending on the region that is biopsied. As a result of this marked heterogeneity, the histological assessment of fibrosis does not have a role for prognosis in BCS when adjusted for liver disease severity [1], and for similar reasons it is difficult to calculate the proportion of patients with cirrhosis at the time of diagnosis.

## 4. BCS in Liver Transplantation

Hepatic venous outflow obstruction can occur in a liver allograft in two main separate settings: (1) recurrent disease in patients transplanted for BCS and (2) de novo.

BCS can recur, and its histological features are similar to those of BCS in native livers. In a multicentre European study of 248 transplanted patients for BCS recurrence, venous thrombosis was observed in 27 patients (portal vein thrombosis in 17 patients, hepatic vein thrombosis in 6 patients, vena cava thrombosis in 5 patients, other sites in 6 patients and thrombosis at two or more sites in 5 patients) [29], despite anticoagulation treatment being given to most patients except 10 patients with protein C or antithrombin III deficiency, who were considered to be cured by LT. In terms of de novo venous outflow obstruction, anastomotic strictures, kinking or thrombosis are rare. The association with the piggyback technique as opposed to caval replacement has been debated [30]. De novo post-transplant hepatic venous outflow obstruction has been divided into four types according to time post-transplant: (1) immediate on reperfusion; (2) acute BCS within 48 h, manifesting with graft dysfunction and ascites and generally due to thrombosis of HV; (3) within 3 months, manifesting with ascites; and (4) chronic [30]. The main histological differential diagnosis in these settings, if a liver biopsy is performed, is due to veno-occlusive changes caused by rejection, including antibody-mediated rejection and drug induced vascular injury [31,32]. In the absence of a significant inflammatory component, features of venous outflow obstruction should be considered as probably related to suboptimal hepatic venous drainage. Chronic antibody-mediated rejection (AMR) is the main consideration in the chronic setting. AMR may be associated with fibrosis without significant inflammatory activity. Correlation with c4d staining and donor-specific antibodies (DSA) is necessary. If a significant infiltrate rich in lymphocytes accompanies signs of venous obstruction, histiocytes and/or plasma cells, then acute or chronic T cell mediated rejection, plasma cell-rich rejection, de novo or recurrent AIH or a drug reaction should be considered [33].

## 5. Conclusions

The histological features of BCS form a complex spectrum which manifests differently in each individual case according to the topographical distribution and chronological evolution of the obliterative insult, the upstream effect of hepatic vascularisation and the consequent parenchymal injury, scarring and remodelling. Liver biopsy is usually not required for the clinical management of patients with BCS. Sampling variation limits its use for prognostication, but it remains critical in those cases (e.g., small hepatic vein BCS or acute liver failure) in which the diagnosis may be uncertain on clinical grounds. Histological differential diagnosis between BCS and other causes of venous outflow obstruction is difficult, and clinico-pathological correlation is critical.

## Figures and Tables

**Figure 1 diagnostics-13-02487-f001:**
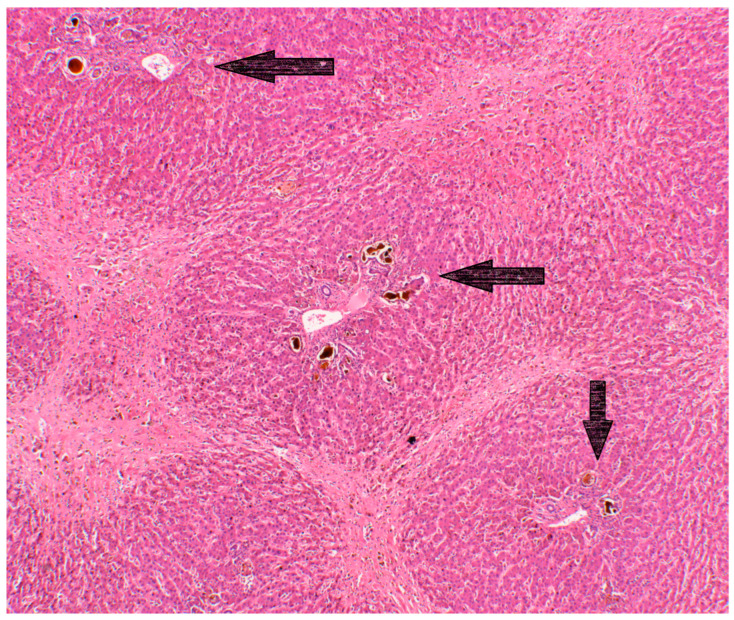
Reverse lobulation and cholestasis in a liver removed at transplantation for BCS. Arrows indicate portal tracts. H&E 40×.

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
