# Peer review of "Histopathology of Budd–Chiari Syndrome"

_diagnostics, 2023, doi:10.3390/diagnostics13152487_

Round 1

Reviewer 1 Report

The review is interesting and well written.

Unfortunately, I cannot see the histological images mentioned in the text. Where are they? They are very important, in a histopathological manuscript. Also, the authors state that biopsy is rarely performed, so they should better explain how these diseases are routinely diagnosed and show some imaging figures.

Additional comment: 

My observations are the same that I reported in the review system. Moreover, now I can see the figure, but not the arrows indicated in the figure legend

none

Author Response

Many thanks for your positive comments. 

I do not know why the arrows do not appear in the submitted version. I have added them again and hope they are visible now. 

I agree with your point about the diagnostic criteria of BCS and imaging features, but this is not a stand-alone review, and these aspects should be addressed by the other more clinical reviews in this special issue dedicated to BCS, hence the reason for not mentioning them in this manuscript focusing on the histopathology of BCS. 

Reviewer 2 Report

The paper is excellent. To minor issues:

Figure 1 can not be opened and viewed.

Some more discussion could be deserved to:

-The rate of patients developing cirrhosis.

-Timing of intervention and the opportunity of early TIPS. 

Author Response

Many thanks for your positive comments. I am resubmitting the figure and hope it will be visible this second time around. In response to your questions:

1) I think it is really difficult to know the rate of cirrhosis in Budd Chiari patients because of the heterogeneity of this disease. To try to work it out based on liver biopsies would be misleading due to sampling error.  Explant livers would be better, but again they show considerable variation. For example, in  the seminal paper by Tanaka and Wanless (Hepatology 1998;27:488-96) on 14 explants and one autopsy liver) , 3 livers were classified as veno-centric cirrhosis (1 with incomplete cirrhosis) 6 livers were classified as mixed veno-centric/veno-portal cirrhosis (3 with incomplete cirrhosis, 2 with parenchymal extinction) and 6 livers were classified as veno-portal cirrhosis (5 with focal parenchymal extinction).

In the paper by Cazals-Hatem (Hepatology 2003;37:510-519) on 17 explants from patients with severe classic BCS , (they referred to Tanaka and Wanless in terms of the assessment of fibrosis) 8 patients show no fibrosis or just perivenular fibrosis (grade 1) in addition to features of venous outflow block, whilst the other nine patients showed a combination of parenchymal macronodules separated by heterogeneously distributed fibrosis ranging from grade 2 (venocentric with bridging) to grade 3 (venoportal bridging) or 4 (venoportal cirrhosis) .  It is the very nature of the disease, and the variable ways in which the histological criteria for the assessment of fibrosis are used which, I think,  argues against trying to calculate reliably the proportion of BCS patients with cirrhosis at diagnosis, as intended in other pathologies.  I have added a short sentence to summarise this point at line 289-290

2) As replied to referee 1, as this paper is part of special issue on BCS, i would think that timing of intervention and role of TIPS should be addressed in the more clinical manuscripts. 

Reviewer 3 Report

The paper is presenting a review of the indications for biopsy and histological findings in budd-chiari-syndrome. 

The review is in my opinion adequately covering the subject, current indications for biopsy and challenges in accurately diagnosing budd chiari induced liver damage are being discussed pointing out the difficulties of distinguishing the variety of ischemic lesions.

The manuscript is in my opinion well structured. The cited references are relevant to the topic. The figures are clear and easy to understand. 

I find the conclusions are consistent with the argumentation; they are in concordance with previous published data and known challenges in the histological interpretation, which needs to be correlated with clinical signs and symptoms.  The conclusions are supported by the listed citations. 

The work is no novelty on the field but summarizes very well histological features caused by hepatic veins obstruction, which is useful considering the complexity of the lesions and distribution pattern.

I suggest a minor revision of the language, with specific focus on the lines: 77-80; 86-87; 274; 311-312;

Author Response

Many thanks for you positive comments.

I have rephrased these sentences accordingly. 
